# Transformers Can Be Translated to First-Order Logic with Majority Quantifiers

## Abstract

Characterizing the implicit structure of the computation within neural networks is a foundational problem in the area of deep learning interpretability. Can their inner decision process be captured symbolically in some familiar logic? We show that any transformer neural network can be translated into an equivalent fixed-size first-order logic formula which may also use majority quantifiers. The idea is to simulate transformers with highly uniform threshold circuits and leverage known theoretical connections between circuits and logic. Our findings also reveal the surprising fact that the entire transformer computation can be reduced merely to the division of two (large) integers. While our results are most pertinent for transformers, they apply equally to a broader class of neural network architectures, namely those with a fixed-depth uniform computation graph made up of standard neural net components, which includes feedforward and convolutional networks.

## 1 Introduction

The incredible success of deep learning models, especially very large language and vision models with tens to hundreds of billions of parameters (Brown et al., 2020; Thoppilan et al., 2022), has come at the cost of increasingly limited understanding of how these models actually work and when they might fail. This raises many concerns, such as around their safe deployment, fairness, and accountability. Is the inner working of such networks fundamentally different from classical algorithms and symbolic systems that we understand better? Or *can their computation be described symbolically using a familiar symbolic formalism?*

We derive the first, to the best of our knowledge, direct connection between a broad class of neural networks and the well-studied classical formalism of first-order logic. Specifically, we show that transformers—and other neural networks with a computation graph that has constant depth and a "repetitive" or uniform structure—implement nothing but fixed-size first-order logic expressions, if the logic is allowed to have *majority* quantifiers (M) in addition to the standard existential (∃) and universal quantifiers (∀). Majority quantifiers simply take a sequence of boolean values and return true if more than half of them are true. The resulting logic formalism is often referred to as $\mathsf{FO}(\mathsf{M})$.

**Theorem 1** (Informal version of Cor. 5.1). *For any neural network $\mathcal{N}$ with a constant-depth computation graph, there is a fixed-size $\mathsf{FO}(\mathsf{M})$ formula $\phi$ equivalent to $\mathcal{N}$.*

This result immediately provides *mechanistic interpretability*—it demonstrates that, at least in principle, the inner decision process of any transformer model can be efficiently translated into a fixed-size formula (with respect to the input length) in a simple, well-defined logic. The output $\mathcal{N}(x)$ of the transformer on any input $x$ is simply the value $\phi(x)$ of this formula. Similar to decision trees, $\mathsf{FO}(\mathsf{M})$ formulae have the property that each sub-expression corresponds to a logical constraint, i.e., a function mapping the input sequence to a truth value. In contrast, the internal modules of a transformer or complex circuit do not satisfy this, as they map between uninterpretable latent spaces. We thus believe that converting transformers to $\mathsf{FO}(\mathsf{M})$ formulae could be leveraged for interpreting their behavior in future work, although a thorough exploration of this idea lies outside the scope of our theoretical contributions in this paper.

Thm. 1 also gives some insight about how to contrast the abilities of transformers and finite-state machines. Classically, the regular languages can be characterized as the languages definable in terms

of monadic second-order logical formulae (Büchi, 1960; Elgot, 1961). We have shown transformers can be simulated by first-order formulae with majority quantifiers. Thus, an advantage transformers have over finite-state machines is the ability to resolve majority quantifiers over their input. However, transformers are not necessarily strictly more powerful than finite-state machines: it is unknown whether second-order monadic quantifiers can be simulated by majority quantifiers.

We derive this connection between transformers and $FO(M)$ by leveraging a key result in circuit complexity theory: the equivalence of $FO(M)$ with highly uniform threshold circuits, specifically with a circuit class known as DLOGTIME-uniform $TC^0$ (Barrington et al., 1990).[1] Our proof builds upon and significantly tightens prior work by Hao et al. (2022), Hahn (2020), Merrill et al. (2022), and Merrill & Sabharwal (2022) on relating specific types of transformer networks to circuit complexity classes. We improve prior analysis on two fronts. First, the class log-uniform $TC^0$ is much tigher than in previous results. Second, in contrast to their work, we obtain a characterization for a fully general model of transformers without limiting assumptions. Our formal model is the first to cover realistic design choices and sizes used in typical transformer implementations.

Specifically, we show that any transformer network with a fixed-depth computation graph can be simulated by a highly uniform class of threshold circuits:[2]

**Theorem 2** (Informal version of Thm. 5). *For any neural network $\mathcal{N}$ with a constant-depth computation graph, there exists a log-uniform $TC^0$ circuit family $\mathcal{C} = \{C_n\}_{n=0}^{\infty}$ such that, for all $x$ of size $n$, $\mathcal{N}(x) = C_n(x)$.*

This result, in addition to helping derive Thm. 1, itself has significant implications. It provides a much tighter class of **transformer-hard problems**, i.e., problems to which any transformer can be efficiently reduced, than previously known. It shows that every problem *complete* for the class of log-uniform $TC^0$ circuits is transformer-hard. Since division (Hesse, 2001; Hesse et al., 2002) is known to be one such problem (Aaronson et al., 2022), this leads to the following rather surprising finding:

**Corollary 2.1.** *For any neural network $\mathcal{N}$ with a constant-depth computation graph and input $x$, the computation of $\mathcal{N}(x)$ can be efficiently reduced to integer division in the following sense: for all $j \in \{1, \ldots, |\mathcal{N}(x)|\}$, there exist efficiently computable (via log-uniform circuits) integers $a_j(x), b_j(x)$, and $i_j$ such that the $j$-th bit of $\mathcal{N}(x)$ equals the $i_j$-th bit of $\lfloor a_j(x)/b_j(x) \rfloor$.*

This again allows us to view transformers from a novel perspective: namely, computing a bit of the output of a transformer with hundreds of billions of parameters can be easily reduced to dividing two integers. Even very large transformers are, in this sense, ultimately simple functions operating at a very large scale.

In summary, our findings shed new light into the inner computation of transformers (and other neural architectures) and its connection to first-order logic. While literature on neuro-symbolic models has often viewed symbolic and neural systems as very different from each other, our results show that the boundaries between these two computational paradigms are not as rigid as they might seem.

**Roadmap.** §2 gives a definition and examples of $FO(M)$, and then introduces relevant background on computation graphs and circuits for our proofs. §3 presents the constructive algorithm to compile computation graph families into threshold circuit families and justifies its correctness. Specifically, we obtain the result that log-uniform computation graphs (with uniform node types) can be compiled into $FO(M)$ expressions. Then, §4 shows that transformers are uniform computation graph families, which implies that they can be compiled into $FO(M)$.

---

[1] For brevity, we will henceforth abbreviate DLOGTIME-uniform as log-uniform.

[2] Conceptually, our results converting transformers to logical formulae and circuits resemble empirical work extracting discrete "subcircuits" from transformers (Elhage et al., 2021), although we use the term circuit in a precise formal sense, rather than to mean any discrete rules summarizing model behavior.

## 2 PRELIMINARIES

### 2.1 FIRST ORDER LOGIC WITH MAJORITY

Transformers can be translated into expressions in $\mathsf{FO(M)}$—what do such expressions look like? We provide a definition of first-order logic with majority as well as some examples of simple languages definable in it.

Informally, $\mathsf{FO(M)}$ is first-order logic extended to also have majority ($\mathsf{M}$) quantifiers. We apply $\mathsf{FO(M)}$ specifically to string inputs, for which quantifiers will range over *indices* into the string, over which various predicates can be applied, as in Barrington et al. (1990).

**Definition 1** ($\mathsf{FO(M)}$ indices)**.** The constants $1$ and $n$ are given as indices in $\mathsf{FO(M)}$. Other indices can be built using addition or subtraction,[3] or can be variables introduced by existential ($\exists$), universal ($\forall$), or majority ($\mathsf{M}$) quantifiers.

**Definition 2** ($\mathsf{FO(M)}$ expressions)**.** For each $\sigma \in \Sigma$ and index $i$, $\mathsf{X}(\sigma, i)$ is a predicate that semantically represents whether the $i$-th token in the input sequence is $\sigma$.[4] Comparing indices with $=$ or $\leq$ also produces a valid expression, as does combining other expressions with the logical connectives $\wedge$ or $\vee$. Finally a new expression can by generated by scoping a quantifier ($\exists, \forall, \mathsf{M}$) with an associated variable over another expression.[5]

A *formula* $\phi$ in $\mathsf{FO(M)}$ is an expression with no free (i.e., unbound) variables. For every input in $\Sigma^*$, it evaluates to a value in $\{0, 1\}$. $\phi$ can thus be viewed as a function from $\Sigma^*$ to $\{0, 1\}$ defining a formal language.

Without loss of generality, we will assume multiple sub-formulae within $\phi$ can be labeled as ordered outputs, thus allowing $\phi$ to be a function from $\Sigma^*$ to $\{0, 1\}^k$ for some fixed constant $k$ bounded above by the number of sub-formulae in $\phi$.

Let $\Sigma = \{\mathtt{a}, \mathtt{b}\}$. To illustrate the formalism, we provide example languages definable in $\mathsf{FO(M)}$:

**Example 1** (Bigram matching)**.** The following $\mathsf{FO(M)}$ expression defines the regular language of strings containing the bigram $\mathtt{ab}$:

$$\exists i \left[ \mathsf{X}(\mathtt{a}, i) \wedge \mathsf{X}(\mathtt{b}, i + 1) \right].$$

**Example 2** (Skip-bigram matching)**.** The following $\mathsf{FO(M)}$ expression defines the language of strings containing the long-distance pattern $\mathtt{a} \ldots \mathtt{b}$, resembling the form of patterns proposed by Elhage et al. (2021) in their analysis of transformer subcircuits:

$$\exists i \left[ \mathsf{X}(\mathtt{b}, i) \wedge \exists j \left[ j \leq i \wedge \mathsf{X}(\mathtt{a}, j) \right] \right].$$

**Example 3** (Majority)**.** The following $\mathsf{FO(M)}$ expression defines the counter language "majority", i.e., the set of strings with more $\mathtt{b}$'s than $\mathtt{a}$'s:

$$\mathsf{M} i \left[ \mathsf{X}(\mathtt{b}, i) \right].$$

Example 1 and Example 2 are definable in first-order logic (without majority), which is a special case of $\mathsf{FO(M)}$. In contrast, Example 3 requires majority predicates (Furst et al., 1981).

### 2.2 COMPUTATION GRAPHS

A *computation graph* over a datatype $\mathbb{D} \subseteq \{0, 1\}^*$ and a countable set of primitive functions $\mathfrak{F} \subseteq \mathbb{D}^* \times \mathbb{D}$ is a directed acyclic graph (DAG) where:

1. Each node is labelled by a *node type*: a function $f \in \mathfrak{F}$ describing the computation of this node.

---

[3]In Barrington et al. (1990) this is not introduced as a primitive, but can be simulated using the $\leq$ predicate.

[4]Barrington et al. (1990) define $\mathsf{X}(i)$ over a binary vocabulary. We generalize this to an arbitrary vocabulary $\Sigma$ by assuming each token is encoded in a one-hot representation. Then $\mathsf{X}(\sigma, i) = \mathsf{X}(|\Sigma| i + s)$ where $s$ is the index of $\sigma$ in the vocabulary.

[5]Sometimes a "bit predicate" is also included in the definition of the formalism. It turns out this is not necessary for $\mathsf{FO(M)}$, as it can be simulated using the other primitives (Barrington et al., 1990).

2. Each edge represents a value $\mathbb{D}$ flowing as output from one node into another node.
3. $\mathfrak{F}$ contains the special symbol input, which designates $k$ nodes as input nodes. We refer to $k$ as the *arity* and assume without loss of generality that nodes $0, \cdots, k-1$ are inputs.[6]
4. A single node is designated the output node. We assume without loss of generality that the output is the node with the largest index.

A computation graph $\mathcal{G}$ of arity $k$ parameterizes a function $f_{\mathcal{G}} : \mathbb{D}^k \to \mathbb{D}$ in the standard way: the input nodes are assigned the input values, and the value of each node is computed (traversing the graph in a bottom-up topological order) a function of the values of its children until the output node receives a value. At this point, the value of the output node is considered the output of the function. It is worth noting that computation graphs can only process inputs of bounded length. To process arbitrary-length inputs, we will need to generalize them to computation graph families (§2.3).

Two important properties for computation graphs are $\mathsf{size}(G)$ (the number of nodes) and $\mathsf{depth}(G)$ (the length of the longest path from an input node to an output node). Additionally, we denote by $\mathsf{arity}(G, i)$ the number of inputs to node $i$.

**Threshold circuits.** A threshold circuit is a special case of a computation graph where $\mathbb{D} = \{0, 1\}$ and $\mathcal{F}$ is the set of threshold functions of the form $\theta_{\leq\Delta}$ and $\theta_{\geq\Delta}$ over $\mathbb{D}^*$, defined as follows: $\theta_{\leq\Delta}(x)$ equals 1 if $\sum_{\sigma \in x} \sigma \leq \Delta$ and 0 otherwise; $\theta_{\geq\Delta}(x)$ is defined analogously. Note that typical boolean circuits with AND, OR, and NOT gates are a special case of threshold gates, as is the IDENTITY logical gate.[7]

Similar to the FO(M) case, we will, without loss of generality, allow nodes with the $k' \geq 1$ largest indices to all be designated as (ordered) output nodes. A threshold circuit with arity $k$ and $k'$ output nodes will thus be a function from $\{0, 1\}^k$ to $\{0, 1\}^{k'}$. This will be convenient when simulating neural network components which often output multiple bits.

We will find it useful to consider threshold circuits as a kind of compilation target for computation graphs: in other words, we will be concerned with simulating computation graphs defined over more complex functions and data types into threshold circuits.

## 2.3 COMPUTATION GRAPH FAMILIES

A computation graph family over $\mathbb{D}$ and $\mathfrak{F}$ is a mapping from $n \in \mathbb{N}$ to a computation graph $G_n$ for processing inputs of size $n$. Thus, $\mathcal{G}$ defines a function from $\mathbb{D}^* \to \mathbb{D}$, where $f_{\mathcal{G}}(x) = f_{|x|}(x)$. Intuitively, computation graph families are useful because they generalize computation graphs to define functions over *unbounded-length* strings as inputs.

**Size, depth, and arity.** For computation graph families, the size and depth become functions of the input length $n$. We define the following functions, which naturally generalize size, depth, and arity to computation graph families:

$$\mathsf{size}_{\mathcal{G}}(n) = \mathsf{size}(G_n)$$
$$\mathsf{depth}_{\mathcal{G}}(n) = \mathsf{depth}(G_n)$$
$$\mathsf{arity}_{\mathcal{G}}(n, i) = \mathsf{arity}(G_n, i).$$

**Uniformity.** $\mathcal{G}$ is an infinite set, so a computation graph family can be more compactly represented by two programs computing the following functions:

1. $\mathsf{node}_{\mathcal{G}}(n, i)$, which returns the type of node $i$ in $G_n$ if $i \leq \mathsf{size}(G_n)$, and $\emptyset$ otherwise.
2. $\mathsf{edge}_{\mathcal{G}}(n, i, j)$, which returns the argument index of $i$ into node $j$ if $G_n$ contains an edge $i \to j$ and $-1$ otherwise. $\mathsf{edge}_{\mathcal{G}}(n, i, j)$ only needs to be defined over $i, j < \mathsf{size}(G_n)$

A pair of algorithms implementing these two functions uniquely specifies a computation graph family, as it enables building the structure of the computation graph $G_n$ for any $n$.

---

[6]By convention in computer science, we let computation graph nodes be zero-indexed.
[7]For more background on threshold circuits, see Merrill & Sabharwal (2022), Merrill et al. (2022), and Arora & Barak (2009).

Table 1: Summary of common notation for computation graph and circuit families.

| Graph | Circuit | Output Range | Description |
|---|---|---|---|
| $i'$ | $i$ | $\mathbb{Z}$ | index of node/gate |
| $\mathsf{node}_{\mathcal{G}}(n, i')$ | $\mathsf{node}_{\mathcal{C}}(n, i)$ | $\widehat{\mathfrak{F}}^9$ | type of node/gate |
| $\mathsf{edge}_{\mathcal{G}}(n, i', j')$ | $\mathsf{edge}_{\mathcal{C}}(n, i, j)$ | $\mathbb{Z}$ | argument number of edge $i \to j$ |
| $\mathsf{size}_{\mathcal{G}}(n)$ | $\mathsf{size}_{\mathcal{C}}(n)$ | $\mathbb{Z}$ | number of nodes/gates in graph/circuit |
| $\mathsf{depth}_{\mathcal{G}}(n)$ | $\mathsf{depth}_{\mathcal{C}}(n)$ | $\mathbb{Z}$ | longest path length in graph/circuit |
| | $\mathsf{bnode}(n, i)$ | $[0, \mathsf{size}_{\mathcal{G}}(n)]$ | index of the block $i'$ containing $i$ |
| | $\mathsf{bstart}(n, i')$ | $[0, \mathsf{size}_{\mathcal{C}}(n)]$ | index of first gate $i$ in block $i'$ |
| | $\mathsf{bsize}(n, i')$ | $\mathbb{Z}$ | size of block $i'$ |

Additionally, it is useful to introduce *uniformity* conditions on computation graph families. Uniform computation graph families (generalizing uniform circuits; cf. Arora & Barak, 2009) are families where $\mathsf{node}_{\mathcal{G}}$ and $\mathsf{edge}_{\mathcal{G}}$ can be computed efficiently, i.e., under some constraints on space or time. In this paper we focus on time-bounded uniformity:

**Definition 3** (Uniformity). A computation graph family $\mathcal{G}$ is $T(n)$-uniform iff each of $\mathsf{node}_{\mathcal{G}}(n, i)$ and $\mathsf{edge}_{\mathcal{G}}(n, i, j)$ can be computed by a deterministic Turing machine in time $T(n)$.

Specifically, we are interested in *log-uniform* computation graph families where $T(n) = \mathrm{O}(\log n)$.

We also introduce a stronger notion of *column uniformity* that captures the highly regular structure of a wide range of commonly used neural network architectures, including the Transformer. This will be useful for tighter bounds by constructing smaller circuits to simulate neural networks:

**Definition 4** (Column uniformity). A computation graph family $\mathcal{G}$ is $T(n)$-column-uniform iff

- There exists a finite sequence of nodes $\{\mathcal{K}_i\}_{0=1}^{\ell-1}$ such that, for $0 \leq i \leq \mathsf{size}_{\mathcal{G}}(n)$, $\mathsf{node}_{\mathcal{G}}(n, i) = \mathcal{K}_{i \bmod \ell}$.[8]
- $\mathsf{edge}_{\mathcal{G}}(n, i, j)$ can be computed by a deterministic Turing machine in time $T(n)$.

We define *log-column-uniform* analogously to log-uniform, i.e., we let $T(n) = \mathrm{O}(\log n)$.

**Threshold circuit families.** Threshold circuit families are families of threshold circuits. We will be interested in simulating computation graph families with threshold circuit families.

## 3   SIMULATING COMPUTATION GRAPHS WITH CIRCUITS

We provide an algorithm that constructs a circuit simulating a given computation graph. Intuitively, the algorithm creates contiguous blocks of circuit gates simulating each node in the computation graph, and routes inputs and outputs between blocks appropriately.

**Block mapping.** This algorithm depends on a *block mapping*, which we define as an implementation of the following three functions:

1. The *block node* $\mathsf{bnode}(n, i)$: the index of the node that gate $i$'s block is simulating.
2. The *block start* $\mathsf{bstart}(n, i')$: the smallest gate index in the block simulating node $i'$.
3. The *block size* $\mathsf{bsize}(n, i')$: the number of gates in the block simulating node $i'$.

Further, we enforce that a valid block mapping must satisfy that, for all $i$, with $i' = \mathsf{bnode}(n, i)$,

$$\mathsf{bstart}(n, i') \leq i < \mathsf{bstart}(n, i') + \mathsf{bsize}(n, i').$$

Now, consider a computation graph $\mathcal{G}$ where the primitive functions are computable by log-uniform threshold circuits. In this case, we can identify each primitive function $f$ with a log-uniform threshold circuit family $\mathcal{T}$ that computes $f$, where the first $\mathsf{arity}_{\mathcal{T}}(n)$ gates are IDENTITY gates reserved for taking input. For such a graph, the $\mathsf{node}_{\mathcal{G}}$ function can be taken to return a symbol identifying a circuit family $\mathcal{T}$.

---

[8]Where $a \bmod b$ represents the remainder when $a$ is divided by $b$.

---

**Algorithm 1** Return the type of gate $i$ in circuit $C_n$.

---

    **procedure** $\mathsf{node}_{\mathcal{C}}(n, i)$
        **if** $i < \mathsf{size}_{\mathcal{G}}(n)$ **then**
            $\mathcal{T} \leftarrow \mathsf{node}_{\mathcal{G}}(n, \mathsf{bnode}(n, i))$
            **return** $\mathsf{node}_{\mathcal{T}}(n, i - \mathsf{bstart}(n, i'))$
        **else return** $\emptyset$

---

**Algorithm 2** Return argument number of edge $i \rightarrow j$ in circuit $C_n$. Return $-1$ if no edge exists.

---

    **procedure** $\mathsf{edge}_{\mathcal{C}}(n, i, j)$
        $i' \leftarrow \mathsf{bnode}(n, i)$
        $j' \leftarrow \mathsf{bnode}(n, j)$
        $s_i \leftarrow \mathsf{bstart}(n, i')$
        $s_j \leftarrow \mathsf{bstart}(n, j')$
        **if** $i' = j'$ **then**                               $\triangleright$ $i, j$ fall within the same node of $G_n$
            $\mathcal{T} \leftarrow \mathsf{node}_{\mathcal{G}}(n, i')$
            **return** $\mathsf{edge}_{\mathcal{T}}(n, i - s_i, j - s_j)$
        **else if** $\mathsf{edge}_{\mathcal{G}}(n, i', j') \geq 0$ **then**           $\triangleright$ $i, j$ fall in nodes of $G_n$ with an edge
            $\mathcal{T}_{i'} \leftarrow \mathsf{node}_{\mathcal{G}}(n, i')$
            $b_i \leftarrow i - (s_i + \mathsf{bsize}(n, i') - p)$        $\triangleright$ adjust $i$ relative to the output bits of $\mathcal{T}_{i'}(n)$
            $b_j \leftarrow j - (s_j + p \cdot \mathsf{edge}_{\mathcal{G}}(n, i', j'))$    $\triangleright$ adjust $j$ relative to the input bits of $\mathcal{T}_{j'}(n)$
            **if** $b_i = b_j < p$ **then return** $j - s_j$
            **else return** $0$
        **else return** $0$

---

In this case, our algorithm requires that, for all $i'$, the block size of $i'$ must match the size of the circuit for the type of block $i'$, i.e., $\mathsf{bsize}(n, i') = \mathsf{size}_{\mathsf{node}_{\mathcal{G}}(n, i')}(n)$.

These properties allow us to meaningfully identify a graph node $i'$ with a block of nodes that will simulate it. This intuition enables us to develop Algorithm 1 and Algorithm 2 for constructing a uniform threshold circuit family from a uniform computation graph family.

**Theorem 3.** *Let $\mathcal{G}$ be a computation graph over a finite set of node types $\mathfrak{F}$, where each $\mathcal{T} \in \mathfrak{F}$ is specified by a log-uniform circuit family. Let $\mathsf{bnode}$, $\mathsf{bstart}$, and $\mathsf{bsize}$ be a valid block mapping in the sense above. Then Algorithm 1 and Algorithm 2 define a circuit family $\mathcal{C}$ such that*

1. *$\mathcal{C}$ and $\mathcal{G}$ compute the same function from $\mathbb{D}_p^* \rightarrow \mathbb{D}_p$, if we consider the final $p$ gates of each $C_i$ to be its output.*
2. *$\mathsf{depth}_{\mathcal{C}}(n) \leq \mathsf{depth}_{\mathcal{G}}(n) \cdot \max_{\mathcal{T}} \mathsf{depth}_{\mathcal{T}}(n)$.*
3. *$\mathsf{size}_{\mathcal{C}}(n) \leq \mathsf{size}_{\mathcal{G}}(n) \cdot \max_{\mathcal{T}} \mathsf{size}_{\mathcal{T}}(n)$.*

*Proof.* Assume without loss of generality that the gates of $\mathcal{C}$ are topologically ordered. We show by induction over circuit gates $j$ (with $j' = \mathsf{bnode}(n, j)$) that:

1. For all $i' < j'$, the last $p$ nodes of block $i'$ store the value of node $i'$.
2. For all $i$ such that $\mathsf{bstart}(n, j') \leq i \leq j$, gate $i$ of $\mathcal{C}$ (as a function of the input nodes of $j'$) computes gate $i - \mathsf{bstart}(n, j')$ of $\mathsf{node}_{\mathcal{G}}(n, j')$.

Base case. We have two circuits with no gates, so the premises are trivially satisfied.

Inductive case. Assume the premises hold up to $j$. We will show they hold for $j + 1$. Let $\mathcal{T} = \mathsf{node}_{\mathcal{G}}(n, j')$. By Premise 1, we know that the last $p$ nodes of block $i'$ store the output of node $i'$, for $i' < j'$. By Algorithm 2, for each $i'$ such that $\mathsf{edge}_{\mathcal{G}}(n, i', j') = a$ with $0 \leq k < \mathsf{arity}_{\mathcal{T}}(n)$, gates $kp$ through $k(p + 1) - 1$ of block $j'$ will copy the final $p$ gates of block $i'$. Thus, the first $k \cdot \mathsf{arity}_{\mathcal{T}}(n)$ gates of block $j'$ store the inputs to node $j'$.

At this point, we use Premise 2 to conclude that the first $j - \mathsf{bstart}(n, j')$ gates of block $j'$ compute the same function as the first $j - \mathsf{bstart}(n, j')$ gates of $\mathcal{T}$ with respect to this input. Thus, we just need to show that gate $j + 1$ is also correct. Within Algorithm 2, we fall in case $i' = j'$, meaning that gate $j + 1$ of block $j'$ gates the same inputs as gate $j + 1$ of $\mathcal{T}$. By Algorithm 1, the type of gate

$j + 1$ in block $j'$ is the type of gate $j + 1$ of $\mathcal{T}$. We conclude that gate $j + 1$ in block $j'$ computes the same function of the input gates as gate $j + 1$ in $\mathcal{T}$. If $j + 1 = \mathsf{bsize}(n, j')$, we further conclude that the final $p$ gates of block $j'$ store the output of node $j'$. $\qquad\square$

Since a transformer computation graph has a constant depth and polynomial size, this theorem allows us to relatively easily recover prior results about hard attention transformers (Hao et al., 2022; Hahn, 2020) and saturated attention transformers (Merrill et al., 2022) using a common framework. All one has to do is show that all individual node types in such transformers can be computed by $\mathsf{AC}^0$ and $\mathsf{TC}^0$ circuits, respectively. We omit the details of these here as later (§4) we will prove a much tighter result showing that all node types in a broader class of neural networks can, in fact, be computed with a much more uniform circuit class.

Formally, let $\mathsf{XC}^0$ denote any family of constant-depth circuits, including $\mathsf{AC}^0$ and $\mathsf{TC}^0$.

**Corollary 3.1.** *Let $\mathcal{G}$ be a computation graph family over a finite set of node types. If $\mathcal{G}$ has constant depth and polynomial size, and every node type in it can be computed by $\mathsf{XC}^0$ circuits, then the function computed by $\mathcal{G}$ is in $\mathsf{XC}^0$.*

### 3.1 Padded Block Mapping for Uniform Computation Graph Families

We now implement a block mapping in the case where $\mathcal{G}$ is a log-uniform computation graph. We analyze the runtime of $\mathsf{node}_\mathcal{C}$ and $\mathsf{edge}_\mathcal{C}$ to show that $\mathcal{C}$ is a log-uniform circuit family.

Our block mapping works by simply padding each block to the maximum size needed to implement any node type in $\mathcal{G}$. We assume that there exists a polynomial function $\mathsf{bsize}(n)$ computable by a Turing machine in $O(\log n)$ time such that, for each node type $\mathcal{T}$, $\mathsf{size}_\mathcal{T}(n) \leq \mathsf{bsize}(n)$. We will later justify this assumption for transformers in §4. Without loss of generality, we can take $\mathsf{bsize}(n)$ to be a power of 2 by padding it to be the smallest power of 2 above its original value.[10]

We then pad each node type circuit $\mathcal{T}$ to have size $\mathsf{bsize}(n)$ by adding unused dummy nodes. We are ready to provide a block mapping:

$$\mathsf{bnode}(n, i) = \lfloor i / \mathsf{bsize}(n) \rfloor$$
$$\mathsf{bstart}(n, i') = i' \cdot \mathsf{bsize}(n)$$
$$\mathsf{bsize}(n, i') = \mathsf{bsize}(n).$$

Since $\mathsf{bsize}(n)$ is a power of 2, $\mathsf{bnode}$ and $\mathsf{bstart}$ are reducible to left and right shifting over $O(\log n)$-bit integers, which can be implemented in $O(\log n)$ time. Thus, these functions are computable in time $O(\log n)$. Since $\mathsf{node}_\mathcal{G}$ and $\mathsf{edge}_\mathcal{G}$ are just calling $\mathsf{bnode}$, $\mathsf{bstart}$, and $\mathsf{bsize}$ with constant overhead, it follows from Thm. 3 that

**Theorem 4.** *Let $\mathcal{G}$ be a log-uniform computation graph over a finite set of node types $\mathfrak{F}$, where each $\mathcal{T} \in \mathfrak{F}$ is specified by a log-uniform threshold circuit family and $\mathsf{size}_\mathcal{T}$ is computable in time $O(\log n)$. Let $\mathsf{bnode}, \mathsf{bstart}, \mathsf{bsize}$ be the padded block mapping w.r.t $\mathfrak{F}$. Then $\mathcal{G}$ can be simulated by a constant-depth, poly-size, log-uniform threshold circuit family via Algorithm 1 and Algorithm 2.*

Following from the equivalence of log-uniform $\mathsf{TC}^0$ and $\mathsf{FO}(\mathsf{M})$, we obtain

**Corollary 4.1.** *Let $\mathcal{G}$ be a log-uniform computation graph over a finite set of node types $\mathfrak{F}$, where each $\mathcal{T} \in \mathfrak{F}$ is specified by a log-uniform threshold circuit family and $\mathsf{size}_\mathcal{T}$ is computable in time DLOGTIME. Let $\mathsf{bnode}, \mathsf{bstart}, \mathsf{bsize}$ be the padded block mapping w.r.t $\mathfrak{F}$. Then $\mathcal{G}$ can be simulated by an $\mathsf{FO}(\mathsf{M})$ formula.*

For both Thm. 4 and Cor. 4.1, the size upper bound from Thm. 3 holds, but we omit it for brevity. The padded block mapping scheme is wasteful in the sense that not every node type needs to receive the maximum block size.

---

[10] $\mathsf{bsize}(n)$ remains $\mathrm{poly}(n)$ because its value at most doubles. It can be computed in $O(\log n)$ time because padding a binary number to a power of 2 simply involves prepending a 1 and converting existing bits to 0's.

## 4 TRANSFORMERS ARE UNIFORM COMPUTATION GRAPH FAMILIES

The last step to make our claim about transformers go through is to justify that the transformer architecture is a log-uniform computation graph family (over node types that can be expressed as log-uniform threshold circuit families). For those familiar with neural networks, it should not be surprising that our notion of a computation graph describes a neural network. In order to justify this rigorously though, we will show that all core neural network components are computable by log-uniform threshold circuit families, and will also derive size and depth bounds for these families.[11]

### 4.1 TRANSFORMERS

We will think of each representation in a transformer as a vector of numbers from $\mathbb{F}_p$, i.e., vectors in $\mathbb{F}_p^d$. In a typical transformer network (Vaswani et al., 2017; Devlin et al., 2019), one may have $p = 16$ or $32$ as a fixed constant determined by the GPU, input length $n = 512$, model dimension $d = 768$, and a potentially larger feedforward dimension $h$. In practice, for newer, larger transformer language models, the precision $p = 16$ (Brown et al., 2020; Zhang et al., 2022).

The model dimension $d$ can be thought of as being roughly the same order of magnitude as $n$. Note that this is much more aligned with practical transformers than the log-precision model of Merrill & Sabharwal (2022), which effectively assumes $dp = \mathrm{O}(\log n)$ and thus applies only for values of $n$ that are substantially larger than what's used in practical transformers (e.g., $dp$ in this example setting is 12288 whereas $\log(n)$ is only 9). Our model and associated result, on the other hand, apply directly to typical implementation choices for several neural net architectures.

The feedforward dimension $h$ can be larger than $d$ in practice (Vaswani et al., 2017; Raffel et al., 2020), but is reasonably still $\mathrm{poly}(n)$. Rather than assuming a specific relationship between $n, d, h$, will derive our characterization of transformer components in terms of these three hyperparameters.

Putting this all together, we can now define a transformer as a computation graph family $\mathcal{N}$. The transformer column $\mathcal{K}_\mathcal{N}$ is a transformer embedding layer followed by $L$ transformer layers, each of which contains a self-attention sublayer and a feedforward sublayer. These sublayers are in turn constructed from affine transformations, activation functions, layer norms, and self-attention modules. We define these primitives (as well as the embedding component) as the functions in $\mathfrak{F}_\mathcal{N}$, and in turn represent the transformer column as a finite sequence of these primitives.

### 4.2 RESULTS

We first justify that the transformer computation graph family is log-uniform, which implies it is log-column-uniform.

**Lemma 1.** *Any transformer computation graph family $\mathcal{N}$ is log-column-uniform.*

*Proof.* We have the column $\mathcal{K}$ by construction. All that remains to show is that $\mathsf{edge}_G$ can be computed in DLOGTIME. All the routing of inputs and outputs between transformer components is fixed within the column besides self-attention. For the fixed routing, we can simply store a finite table for each $i', j' \mod |\mathcal{K}|$ and look up the answer to answer $\mathsf{edge}_G(n, i', j')$. For routing into a self-attention component $j'$ at sequence position $t_{j'}$, let $i'$ is the output of the previous layer at position $t_{i'}$. For full attention, we simply return $t_{i'}$. For causally masked attention, we return $t_{i'}$ if $t_{i'} < t_{j'}$ and 0 otherwise. Either way, this logic is clearly possible in DLOGTIME. Thus, we conclude that the transformer computation graph family is column-uniform. □

Next, we need to justify that each neural net component used as a node in the transformer computation graph is uniform and that its size is computable in DLOGTIME.

**Lemma 2.** *Assume $d, h, |\Sigma| \leq \mathrm{poly}(n)$. Each $f \in \mathfrak{F}_\mathcal{N}$ is computable by a log-uniform threshold circuit $\mathcal{T}$ with $\mathsf{size}_\mathcal{T}$ computable by a deterministic Turing machine in time $\mathrm{O}(\log n)$.*

---

[11]While we provide this rigorous justification for transformers, similar arguments can also be made for a broader class of neural networks made up of standard neural net components, which includes feedforward and convolutional networks.

*Proof.* Out of all the transformer components (Vaswani et al., 2017), activation functions and residual connections are completely "local" (cf. §A), meaning they have fixed-depth, fixed-size circuits as a function of $n$, $d$, and $|\Sigma|$. Thus, these operations are trivially uniform.

It remains to be shown that the other "global" components (embeddings, affine transformations, layer norm, and self attention) can be computed by uniform threshold circuit families with size computable in DLOGTIME. We do this in §B by reducing each component to summation with uniform local functions. This allows us to derive the following:

1. By Lem. 8, embedding the input token and position is computable by a uniform threshold circuit family of size at most $O(|\Sigma|d + n)$.
2. By Lem. 9, each affine transformation is computable by a uniform threshold circuit family of size at most $O(\max\{d^2 \log d, dh \log h\})$.
3. by Lem. 10, the layer norm is computable by a uniform threshold circuit family of size at most $O(d \log d)$.
4. By Lem. 11, self attention is computable by a uniform threshold circuit family of size at most $O(n \log n)$.

For each component type $\mathcal{T}$, we pick a function $T(n)$ that is both larger than the size bound for $\mathcal{T}$ and computable by a deterministic Turing machine in $O(\log n)$ time. We pad $\mathcal{T}$ with unused nodes to make $\mathsf{size}_{\mathcal{T}}(n) = T(n)$.

We conclude that each $f \in \mathfrak{F}_{\mathcal{N}}$ is computable by a log-uniform threshold circuit $\mathcal{T}$ with $\mathsf{size}_{\mathcal{T}}$ computable by a deterministic Turing machine in time $O(\log n)$. $\qquad\square$

This column uniformity implies standard uniformity, and that the overall size is the column size times $n$. We can now directly obtain the following applying Thm. 4 to transformers:

**Theorem 5.** *For $d, h, |\Sigma| \leq \mathrm{poly}(n)$, let $\mathcal{N}$ be a transformer computation graph over vocabulary $\Sigma$ with model dimension $d$ and feedforward dimension $h$. Then the function $\mathcal{N}$ defines over $\Sigma^*$ is computable by a constant-depth threshold circuit family $\mathcal{C}$ with size at most*

$$n \cdot O\left(|\Sigma|d + d^2 \log d + dh \log h + n \log n\right).$$

The $n$ comes from the number of columns, and the other factor comes from the size of a column, where the terms follow from the sizes of transformer components derived in Lem. 2. Practically, in an overparameterized transformer, $|\Sigma|$ or $h$ is often the largest hyperparameter. If so, we would expect the terms $|\Sigma|d$ and $dh \log h$ to dominate the size of the threshold circuit. These correspond to the the circuit gates simulating the embedding layer and the feedforward sublayers, respectively.

Analogously to Cor. 4.1, we show that transformers can be translated to equivalent $\mathsf{FO}(\mathsf{M})$ formulae:

**Corollary 5.1.** *For $d, h, |\Sigma| \leq \mathrm{poly}(n)$, let $\mathcal{N}$ be a transformer computation graph over vocabulary $\Sigma$ with model dimension $d$ and feedforward dimension $h$. Then $\mathcal{N}$ can be simulated by an $\mathsf{FO}(\mathsf{M})$ formula.*

## 5 CONCLUSION

We proved that any transformer model can be efficiently translated into a fixed-size $\mathsf{FO}(\mathsf{M})$ logic formula that computes the same function as the transformer (for all input lengths). This result comes by first simulating a transformer with a highly uniform threshold circuit family, and then leveraging the established equivalence of log-uniform circuits and $\mathsf{FO}(\mathsf{M})$. Transformers and other neural nets are often discussed in contrast with symbolic models based on logical formalisms (Garnelo & Shanahan, 2019)—an immediate implication of our result is that it is possible to express the inner workings of transformers also in a simple logic, challenging the premise of a rigid division between symbolic and neural models.

It would be an interesting future direction to implement our efficient algorithms to convert transformers to $\mathsf{FO}(\mathsf{M})$ formulae, and leverage this logical representation for increased interpretability of the model. Extending our results to recursive architectures such as RNNs also remains an interesting open problem.

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

# A    LOCAL AND GLOBAL FUNCTIONS

## A.1    COMPOSITION

To justify our modular decomposition of transformers, we first show that the composition of $O(\log n)$-uniform circuit families is also a $O(\log n)$-uniform circuit family, assuming the sizes of the input families in the composition are computable in DLOGTIME.

**Lemma 3** (Composition). *For a fixed $1 \leq i \leq d$, let $f_i : \mathbb{D}^k \to \mathbb{D}$ be computable by a $O(\log n)$-uniform computation graph $\mathcal{G}_i$ over $\mathbb{D}, \mathfrak{F}$ with $\mathsf{size}_{\mathcal{G}_i}(n)$ computable in DLOGTIME. Let $g : \mathbb{D}^d \to \mathbb{D}$ be a $O(\log n)$-uniform computation graph over $\mathbb{D}, \mathfrak{F}$. Then, for all $x \in \mathbb{D}^k$, the following is computable by a DLOGTIME uniform computation graph $\mathcal{G}^*$ over $\mathbb{D}, \mathfrak{F}$:*

$$f \circ g(x) = g\left(f_1(x), \cdots, f_d(x)\right),$$

*where the size and depth of $\mathcal{G}^*$ are at most additive in the size of circuits computing each $f_i$ and $g$.*

*Proof.* Let $\mathcal{G}^*$ be the computation graph computing the composed function. We first compute the size of each circuit $\mathcal{G}_i$ in DLOGTIME and call that $s_i$. To answer $\mathsf{node}_{\mathcal{G}^*}$ and $\mathsf{edge}_{\mathcal{G}^*}$ queries, we can then partition the indices between 0 and $S = \sum_{i=1}^d s_i$ to represent different $f_i$'s — this requires only a finite number of cases. We use indices $\geq S$ to represent the nodes of $g$, and route the output (last $p$ nodes) of each $f_i$ to $g$ as input. □

## A.2    LOCAL FUNCTIONS

**Lemma 4** (Extended from Hao et al., 2022, Lem. 3). *Any function $f : \{0,1\}^m \to \{0,1\}^p$ can be computed by a depth-3 circuit of size at most $2^m + m + p$.*

*Proof.* Similar to Hao et al. (2022), we construct a circuit using a DNF representation of $f$, except that we use a combined DNF representation for all output bits of $f$. The DNF formula has at most $2^m$ terms. The circuit has a NOT gate for each input bit, an AND gate for each DNF term, and, for each of the $p$ output bits, an OR gate combining the outputs of those AND gates (i.e., DNF terms) for which that bit is 1. □

**Corollary 5.2** (Local functions). *Any $p$-precision arithmetic function of $k$ numbers can be computed by a depth-3 circuit of size $2^{kp} + kp + p$.*

## A.3    GLOBAL FUNCTIONS

Two lemmas of Kayal (2015) on addition with threshold circuits will be useful for building circuit families to add floats.

**Lemma 5** (Extended from Kayal, 2015, Thm. 5.8). *Any symmetric function $\{0,1\}^* \to \{0,1\}$ can be computed by a $O(\log n)$-uniform threshold circuit family of depth 2 and size $O(n)$ (where $n$ is the input bitstring length).*

*Proof.* The correctness of the circuit family is shown by Kayal (2015); uniformity follows from the constructive algorithm in their proof. □

**Lemma 6** (Extended from Kayal, 2015, Thm. 5.9). *The sum of $k$ integers of $k$ bits each can be computed by a $O(\log n)$-uniform constant-depth threshold circuit family of size $O(k^3)$.*

*Proof.* The correctness of the circuit family is shown by Kayal (2015); uniformity follows from the constructive algorithm in their proof. □

We can use these lemmas to analyze adding $k$ floats of precision $p$, which is our notion of global computation in the transformer.

**Lemma 7.** *The sum of $k$ floats of $p$ bits each can be computed by a $O(\log n)$-uniform constant-depth threshold circuit family of size $O(k \log k)$, with a constant depending on precision $p$.*

*Proof.* The proof procedes by constructing $O(\log n)$-uniform circuit families to carry out the following computations, where each threshold circuit family $\mathcal{C}$ has $\text{size}_{\mathcal{C}}$ computable in DLOGTIME. The composition of these functions gives the correct sum truncated to precision $p$, so thus by Lem. 3 the sum is computable by a $O(\log n)$-uniform circuit family whose size is the sum of the constituent circuit sizes. All that remains is to justify the circuit family size for each step.

1. Let $p' = p/2 + \exp(p/2)$. We first map the $k$ $p$-bit floats to $k$ $p'$-bit integers representing the same value.

2. Sum the bits in each place value across all $k$ integers, yielding $p'$ integers of size $\log k$.

3. Shift each place value integer so that they line up semantically, and add. This corresponds to adding $p'$ integers of size at most $\log k + p'$ to produce a new integer $s$.

4. Finally, return the first $p$ non-zero leading bits of $s$ as the mantissa, and the number of remaining bits (at most $p' - p$) as the exponent.

Step 1: Floats to ints. This involves local functions applied to each number, so its size is $O(k)$.

Step 2: Summing across place values. Observe that summing $k$ bits across each place value is a symmetric function with $\log k$ output bits. Thus, by Lem. 5, we have a a $O(\log n)$-uniform constant-depth threshold circuit family of size $O(k)$ that computes each of the $\log k$ output bits for each place value up to $p'$. Summing over all $\log k$ bits and a constant number of place values, the size of the overall circuit for this step is $O(k \log k)$.

Step 3: Addition. We ignore the size of local functions, since they have a fixed-size circuit with respect to $k$ and thus take $O(k)$ size when applied elementwise. Adding $p'$ integers of size at most $\log k + p'$ can be done by a constant-depth threshold circuit family of size $2(\log k + p')^3 = O(\log^3 k)$ using Lem. 6.

Step 4: Ints to floats. This involves local functions applied to the final sum, so its size is $O(1)$.

Summing the sizes of each step yields an overall size of

$$O(k) + O(k \log k) + O(\log^3 k) + O(1) = O(k \log k).$$

$\square$

# B   TRANSFORMER COMPONENTS

## B.1   TRANSFORMER EMBEDDINGS

The initial layer of the transformer embeds a token $\sigma$ from a vocabulary $\Sigma$, as well as their position from $1 \leq t \leq n$, into a vector $\mathbf{e}(\sigma, t)$. Let $\mathbf{V}$ be an embedding matrix of size $|\Sigma| \times d$ where each row represents the embedding for some $\sigma$. Then,

$$\mathbf{e}(\sigma, t) = \mathbf{v}_{\sigma} + f(t).$$

**Lemma 8.** *The transformer embedding layer can be computed by a constant-depth log-uniform threshold circuit family of size* $O(|\Sigma|d + n)$.

*Proof.* The first term of the embedding mechanism can be implemented with a lookup table encoded as a circuit of size $O(|\Sigma|d)$. To build the circuit for the second term at position $t$, we first construct the constant $t$ as a $\log n$-bit integer. We then apply Lem. 4 to compute $f(t)$ in time

$$2^{\log n} + \log n + d = O(n + d).$$

Thus, the full embedding layer can be computed as a function of an input string $w \in \Sigma^*$ by a constant-depth log-uniform circuit family of size $O(|\Sigma|d + n)$. $\square$

## B.2 AFFINE TRANSFORMATIONS

Affine transformations form the building block of neural network layers. In a transformer, they occur in the both the self-attention sublayer as key, query, value, and output transformations, and in the feedforward sublayer in standard neural network layers.

**Lemma 9.** *An affine transformation from $\mathbb{F}_p^k$ to $\mathbb{F}_p^{k'}$ can be computed by a $\mathrm{O}(\log n)$-uniform circuit family of constant depth and size $\mathrm{O}(k'k \log k)$.*

*Proof.* For the transformation $\mathbf{W}\mathbf{x} + \mathbf{b}$, first compute each $\mathbf{w}_i \odot \mathbf{x}$ in parallel, where $\odot$ represents elementwise multiplication. By applying Cor. 5.2 in parallel $kk'$ times, this can be done with a circuit of depth 3 and size $kk'(2^{2p} + 3p) = \mathrm{O}(kk')$. We also have $k'$ circuits, each corresponding to an index of the output, that compute the sum over the $k$ entries of each $\mathbf{w}_i \odot \mathbf{x}$. By Lem. 3, the composition of these two functions is computable by a $\mathrm{O}(\log n)$-uniform circuit family of constant depth and size $\mathrm{O}(kk') + k'\mathrm{O}(k \log k) = \mathrm{O}(kk' \log k)$. □

The transformer's key, query, value, and output affine transformations with $k, k' \leq d$. The transformer also has affine transformations in the feedforward subnetwork with $k, k' \leq \max\{d, h\}$. So, the size of an affine transformation circuit in the transformer is at most

$$\mathrm{O}(\max\{d^2 \log d, dh \log h, hd \log d\}) = \mathrm{O}(\max\{d^2 \log d, dh \log h\}).$$

## B.3 LAYER NORM

The layer norm is applied between sublayers in the transformer. Let $\mu = (1/d) \sum_{i=1}^d x_i$. The layer norm $\mathbf{y} \in \mathbb{F}_p^d$ of a vector $\mathbf{x} \in \mathbb{F}_p^d$ is computed, for scalars $a, b \in \mathbb{F}_p$,

$$\mathbf{y} = a \left( \frac{\mathbf{x} - \mu}{\|\mathbf{x} - \mu\|} \right) + b.$$

**Lemma 10.** *The layer norm over a vector of size $d$ can be computed by a uniform threshold circuit of constant depth and size $\mathrm{O}(d \log d)$.*

*Proof.* First compute $d$ using summation over the constant term 1 from 1 to $d$. This summation has a threshold circuit size of $\mathrm{O}(d \log d)$ by Lem. 7. Then compute $\mu$ using a similar sum circuit, composed with the circuit computing $d$. Finally, we need to compute $\|\mathbf{x} - \mu\|$ which can be reduced to a single summation by local functions (squaring and dividing) with $\mathrm{O}(d)$ overhead. Applying Lem. 3 to compose these functions yields a constant-depth uniform threshold circuit of size $\mathrm{O}(d \log d)$. □

## B.4 SELF ATTENTION

Define $d_k, d_v \leq d$. The self attention mechanism takes a query $\mathbf{q} \in \mathbb{F}_p^{d_k}$, $n$ keys $\mathbf{k}_i \in \mathbb{F}_p^{d_k}$, and $n$ values $\mathbf{v}_i \in \mathbb{F}_p^{d_v}$. It returns a new vector $\mathbf{y}$ of size $d_v$ computed, for $1 \leq i \leq n$,

$$\mathbf{y} = \sum_{i=1}^n (\alpha_i/Z)\mathbf{v}_i.$$

where $\alpha_i = \exp(\mathbf{q} \cdot \mathbf{k}_i)$ and $Z = \sum_{j=1}^n \alpha_j$.

**Lemma 11.** *Self attention is computable by a $\mathrm{O}(\log n)$-uniform threshold circuit of constant depth and size $\mathrm{O}(n \log n)$.*

*Proof.* We can compute each $\alpha_i$ using only local functions. Computing $Z$ takes one sum circuit over $n$ scalars. For $\mathbf{y}$, we just need an addition summation circuit over $n$ terms and some local functions to compute each term. We can apply Lem. 3 to conclude that the composition of these circuits is computable by a uniform threshold circuit of constant depth and size $\mathrm{O}(n \log n)$. □

