# OpenReview forum: "Transformers Implement First-Order Logic with Majority Quantifiers"
_ICLR.cc/2023/Conference — Submitted to ICLR 2023_

### Official Review · Reviewer_51Kn · 2022-10-25

**Confidence:** 3
**Correctness:** 4
**Technical Novelty And Significance:** 3
**Empirical Novelty And Significance:** Not applicable
**Recommendation:** 8

**Clarity, Quality, Novelty And Reproducibility:**

The paper is clearly structured overall. However, I find some part of the writing confusing, e.g. especially the proof for Thm 3.
Here are some other notes:
- Since first-order logic is never explicitly used and only appears by the equivalence shown in Barrington 90, I'd put Sec 2.1 in the appendix, and instead move Lem 5,6 to the main paper (e.g. right after threshold circuit is defined in Sec 2.2) since they provide why threshold gate matters.
- For clarity, it would be helpful to explain why "column uniformity" is called "column", e.g. "a column of nodes" as mentioned in the appendix.
- Seemingly inconsistent notations:
    - a computation graph is denoted both by $\mathcal{G}$ (e.g. when used in the last paragraph of page 3) and by $G$.
    - $\mathcal{G}$ is used to denoted both a computation graph and a computation graph family.
- Some typos
    - the first bullet point of Def 4
    - Alg 1, the return in the first case: $i'$ is not defined in the scope of this function; should it be $i' = \mathsf{bnode}(n, i)$?
    - Sec 4.1, last sentence in the 3rd paragraph.
    - Proof for Lem 1: 5th line in the proof.




**Strength And Weaknesses:**

Besides the main result that transformers can be compiled into FO(M), there are some interesting implications:
- If we treat transformer as a complexity class, then one implication of the result is that any problem that is complete for log-uniform $TC^0$ will be "transformer-hard". One example is division, which means transformer computation can be reduced to integer division.
- This opens up the potential to connecting to logical formula, though the exact connection is left as future work.

The proof is short and clean, and even though there's not much new/complex proof techniques, the connection to the right prior work is clever. The main idea is that:
- Thm 3, 4: a log-uniform computation graph $\mathcal{G}$ can be compiled into a circuit in log-uniform $TC^0$.
  - Cor 4.1: by the equivalence between log-uniform $TC^0$ and FO(M) shown in Barrington et al.1990, $\mathcal{G}$ can also be compiled into a FO(M).
- Thm 5, Cor 5.1: a poly-sized transformer can be compiled into a log-uniform $TC^0$ circuit and hence a FO(M) formula, by showing that every transformer component is computable by a poly-sized uniform threshold circuit family.

The connection between circuits (binary valued) and finite precision float-computation is given in Cor 5.2 (Appendix A.2).

I don't have major complaints about the paper; here are some minor comments/clarifications:
- the claim that the results help blur the boundary between symbolic and neural methods is interesting but also a bit of a stretch; I'd be more convinced if there's an exact compilation into FO(M) is provided, which is however currently left as future work.
- footnote 4 on page 3: $X(\sigma, i) = X(i + s)$: I don't follow this; e.g. why the sum?
- Before Example 1, "Let $\Sigma = \{a,b\}$": does this definition of $\Sigma$ apply to the 3 examples only, or throughout the paper?
- Table 1: what's the superscript 9 in $\mathfrak{F}^9$?
- Alg 1: I'm missing something here: is this a recursive function? If yes, wouldn't this recursion always return $\varnothing$, since it's the only base case?
- Besides Lem 5 & 6, are there other places that use properties specific to _threshold_ circuit?
- Cor 3.1: not sure why this is called a corollary (of Thm 3?), since this is true by the definition of $XC^0$.
- Proof of Thm 3: what is $a$ (assigning to $\mathsf{edge}_{\mathcal{G}}(n, i', j')$)?

**Summary Of The Paper:**

This paper shows that transformers can be represented by FO(M) formula (i.e. first-order logic with majority quantifiers), by showing that transformers can be compiled into a circuit in log-uniform $TC^0$, and then using the equivalence between log-uniform $TC^0$ and FO(M) provided in Barrington et al.1990.

While the connection between transformers and circuits is not new, the proof in this paper is clean and subsumes the results in prior work that 1) hard-attention can only recognize $AC^0$ (e.g. Hahn 20, Hao et al. 22) and 2) saturated attention can only recognize $TC^0$ (e.g. Merrill et al. 22).

**Summary Of The Review:**

This is a cute paper with interesting results and clean proof ideas, extending the line of work connecting transformers to circuits, and building new connection to formal logic (though the connection to logic is via Barrington 90 and not novel).
The paper is well structured overall with some writing details to be improved.

---

> ### Author Response · Authors · 2022-11-18
> **Response to Reviewer 51Kn**
>
> Thank you for your encouraging review!
>
> Regarding a constructive mapping from transformers to $\mathsf{FO}(\mathsf M)$, we have some new insights on this, and believe it is very possible. The idea is to express Algorithms 1 and 2 not as log-time Turing machines, but instead as FO logic formulae themselves (a formula for “is T the type of node i?” and a formula for “is there an edge between nodes i,j?”). We can then apply a known construction to build a formula computing the threshold circuit family out of these primitives. This would constitute a significant effort beyond the scope of the current work, but we plan to explore this in the near future. Thank you very much for the suggestion!
>
> To clarify, calligraphic $\mathcal G$ refers to a computation graph family (set of computation graphs for different $n$), whereas $G$ or $G_n$ refers to a computation graph. Regarding $X(i, \sigma)$, we are saying that if each input token $i$ is encoded by a one-hot vector, then $X(i, \sigma)$ can be computed by taking the the $\sigma$-th bit of the vector. There is a small typo that we have fixed in the revised version: it should read $\mathsf X(i, \sigma) = \mathsf X(i \cdot |\Sigma| + s)$.

---

### Official Review · Reviewer_hzSU · 2022-10-25

**Confidence:** 4
**Correctness:** 3
**Technical Novelty And Significance:** 1
**Empirical Novelty And Significance:** Not applicable
**Recommendation:** 3

**Clarity, Quality, Novelty And Reproducibility:**

The presentation is not always clear and could be improved significantly. The novelty is limited, since, in my understanding, this result does not allow us to conclude much about what transformers can uniformly compute. Some statements of the paper are just imported from well-known results, i.e., division is TC0-hard, but authors present these to make their result sound more interesting/surprising: "This again allows us to view transformers from a novel perspective: namely, computing a bit of the output of a transformer with hundreds of billions of parameters can be easily reduced to dividing two integers."

**Strength And Weaknesses:**

Strengths:

- There are big gaps in our understanding of large language models, and so a formal, logic-based analysis of transformers is interesting and  promising.

 Weaknesses:

- The connection between circuit classes and neural networks is not new as claimed in the paper (and neither is the connection to logic) -  see, e.g., "Parberry, Circuit Complexity and Neural Networks, 1994". The difference is, of course, in analysing the modern neural network architectures with different assumptions (e.g., nonlinearities, real-values, etc), but it is by no means a new idea, as claimed in the paper: "We derive the first, to the best of our knowledge, direct connection between a broad class of neural networks and the well-studied classical formalism of first-order logic." I strongly advise the authors to revisit the classical as well as recent literature on these topics.

- Informally, the result of this paper is to show that any transformer model can be represented by a formula in FO(M). This follows from the fact that each computation graph of a transformer model can be represented by a DLOGTIME-uniform TC0 circuit, and each such circuit corresponds to a formula in FO(M). In my opinion, this is not a very strong result and I could not spot anything specific about transformers in their results.

- The presented result does not tell us what class of functions transformers can capture. It only gives an upper bound on this (and details are not clear to me) in showing that the corresponding computation graphs can always be mapped to FO(M) - how about the other direction? Is it the case that any formula in FO(M) can be *uniformly* captured by transformers?

- There are many related results for neural networks, see, e.g., "Barcelo et al., The Logical Expressiveness of Graph Neural Networks, ICLR 2020": they show that a particular class of graph neural networks can *uniformly* capture 2-variable first-order logic with counting quantifiers in the sense that for any formula in this logic, there is a corresponding graph neural network which precisely computes the function that this formula encodes.



**Summary Of The Paper:**

This paper analyses transformers through the lens of logic, where the goal is explain the computation graph of a transformer (and of some other models) in terms of a formal language. The computation graph of most neural networks with a fixed number of layers is easily seen to be a directed acyclic graph. Hence, it is possible to characterise neural networks in terms of depth-bounded circuits ad the corresponding families AC0 or TC0. Using the well-known connection between (DLOGTIME-uniform)-TC0 and FO(M) - first-order logic with majority quantifiers- the authors show that any computation graph of a transformer can be represented in FO(M).

**Summary Of The Review:**

Given my concerns in the review, I recommend a reject.

---

> ### Author Response · Authors · 2022-11-18
> **Response to Reviewer hzSU**
>
> Thank you for your review! We'd like to address your core comments.
>
> **Classical papers:** We will make sure to incorporate references to classical papers connecting neural networks and circuit complexity. We note, however, that the old results apply to feedforward neural networks processing fixed-size inputs. Further, they do not provide a link to logic, which is one of our key messages. As you say, our results are new in that they apply to transformers processing variable-length sequences. While this may appear to be a small difference, our findings are much more refined, quite a bit more technical, and significantly more important in establishing new connections between modern neural network architectures and logic as well as well-understood problems like integer division.
>
> **Nothing specific about transformers:**  We state our results at a higher level of generality than just proving them for the transformer architecture. The key formal property of transformers that enables the result is that the transformer computation graph has constant depth with respect to standard vector operations allowed in a neural network.
>
> **Some statements are just imported from well-known results:** Our intent in mentioning well-known results about problems outside log-time uniform TC0 is to show how our result establishes a connection between transformers and existing theory. We believe it is a valuable contribution to do the work of connecting transformers to log-time uniform TC0, so that knowledge about TC0 can be imported for understanding transformers. For instance, our findings allow us to observe that the transformer computation can be easily reduced to the division of two integers. Such observations provide new perspectives that, we believe, are valuable to spell out.

---

### Official Review · Reviewer_bNHi · 2022-10-28

**Confidence:** 2
**Correctness:** 3
**Technical Novelty And Significance:** 4
**Empirical Novelty And Significance:** 4
**Recommendation:** 6

**Clarity, Quality, Novelty And Reproducibility:**

clarity: the authord write well, but the amount of background knowledge required to just follow the paper is a lot. On the other hand, given that reading the Apps is required anyway, maybe there could be room for an example
:nov: looks novel, but I am not an expert.
repro: ibd


**Strength And Weaknesses:**

Strengths:
It looks like an interesting result.
Weaknesses:
It is unclear how useful a result it is.gooh

**Summary Of The Paper:**

The paper establishes a connection between a form of first-order-logic and a significant class of NNs.

**Summary Of The Review:**

The result looks important, but my score reflects my limitations in evaluating the work.

Q; where dies FO(M) come from? I could see no reference in S2
Q You seem to suggest it is more expressive than FO? I suppose the difference is in the ability to count solutions.Also. I assume correctness hold.
Q:In my weak understanding, you graph->logic transformation and may generate complex Graphs and formulae. Would this be  such a great step in understanbility?

---

> ### Author Response · Authors · 2022-11-18
> **Response to Reviewer bNHi**
>
> Thank you for your comments!
>
> We will add some details about the logic class $\mathsf{FO}(\mathsf{M})$ and its significance in the known literature. Briefly, it is one of the smallest yet reasonably expressive and intuitive complexity classes that includes basic arithmetic (addition, multiplication, division). It is similarly one of the smallest yet reasonably expressive complexity classes to implement dot products over linear-sized vectors, which is a key part of computing a neural network.
>
> **Understandability:**  While we agree that logical expressions with a huge size can be difficult to interpret, we also believe that first-order logic formulae with majority are structurally more understandable than a large transformer or circuit graph. The principal reason for this is that, even in a very large logical expression, the quantified variables in the expression always have a straightforward interpretation as positions/tokens in the input string. The predicates in the expression thus directly express properties of the input sequence, rather than functions of uninterpretable hidden states like in a neural net. This property, that each predicate directly expresses a constraint on the input, makes the formulae representation potentially inefficient for inference, but more useful as a interpretability and debugging tool. We have added the following discussion in the paper clarifying this:
>
> > Similar to decision trees, FO(M) formulae have the property that each sub-expression corresponds to a logical constraint, i.e., a function mapping the input sequence to a truth value. In contrast, the internal modules of a transformer or complex circuit do not satisfy this, as they map between uninterpretable latent spaces. We thus believe that converting transformers to FO(M) formulae could be leveraged for interpreting their behavior in future work, although a thorough exploration of this idea lies outside the scope of our theoretical contributions in this paper.
>
> We also note that the equivalence to first-order logic has value besides interpretability, in that it gives us problems like integer division that any transformer can be efficiently reduced to.

---

### Official Review · Reviewer_kJX7 · 2022-11-02

**Confidence:** 3
**Correctness:** 3
**Technical Novelty And Significance:** 2
**Empirical Novelty And Significance:** Not applicable
**Recommendation:** 5

**Clarity, Quality, Novelty And Reproducibility:**

Several recent works have shown similar reductions from Transformers to threshold circuits. What's new seems to be pointing out the equivalence between threshold circuits and first-order logic with majority quantifiers. However, this observation does not provide any new insight for the reduction itself.  The translation/reduction process is a bit involved, but it seems to be a simple adaption of recently published work. The resource-bound analysis may be interesting, which is also straightforward (i.e., composing standard sub-components).


**Strength And Weaknesses:**

Strengths:
- although being fairly theoretical, the paper is well-written and properly introduces all the necessary backgrounds
- the limit (or potential of neural network) models is justified formally and quantitatively, e.g., detailed bound analyses are provided.

Weaknesses:
- the idea is incremental compared to several recent works, especially MSS22
- given this is a theory paper, conclusions would be excepted to be rigorous. However, some conclusions are either informal or misleading, including the title. Transformers can be translated (or compiled) into a class of threshold circuits TC0, which is equivalent to FO(M). This does not imply that TC0 or FO(M) can be implemented in Transformers.  Similarly, the fact that integer division is reducible to TC0 does not imply that Transformers is equivalent to some integer division.


[MSS22] Saturated Transformers are Constant-Depth Threshold Circuits. William Merrill, Ashish Sabharwal, and Noah A. Smith. Transactions of the Association for Computational Linguistics, 2022.

**Summary Of The Paper:**

This paper presents a simple translation from computation graphs with a constant depth, which describe many neural network models including Transformers, into a class of threshold circuits, i.e., DLOGTIME-uniform $TC^0$, or equivalently, expressions of first-order logic with majority quantifier. As a result, the limit of threshold circuits implies the limit of computation graphs like Transformers.



**Summary Of The Review:**

The reduced threshold circuits closely resemble the topological structure of computation graphs. Given the computation graph has a finite depth, the reduced circuits also have a finite depth. Either conclusion like this made in this paper sound obvious, or there is something significant I do not really understand. To me, it is like claiming all modern computers are just finite-state machines (or regular expressions) because the size of the memory or hard disk can only be finite.

---

> ### Author Response · Authors · 2022-11-18
> **Response to Reviewer kJX7**
>
> Thank you for your review! We’d like to address your main concerns about 1) the novelty, 2) the title, and 3) the integer division claim.
>
> ### Novelty
>
> The result in the paper you cite (MSS22) is substantially limited and weaker than ours on two fronts: it shows transformers can be simulated by **non-uniform** TC0 circuits, and it only considers a very simplified model of attention. In contrast, our result considers a realistic model of transformers and shows that it can be simulated by **log-time uniform** TC0 circuits.
>
> This new result may appear incremental, but it is much more refined, quite a bit more technical, and significantly more important than the non-uniform one.
>
> Specifically, log-time uniformity is necessary for the equivalence of TC0 and FO(M), and thus to conclude that transformers can be simulated by logical formulae. Further, division and other log-time uniform TC0-complete problems are transformer-hard only if transformers can be simulated by log-time uniform TC0. These new intriguing connections to well-known problems and formalisms do not follow from the MSS22 paper.
> In addition to the significance of the log-time uniform result compared to the non-uniform one, we also point out that the proof technique is different, delicate, and more involved, as it requires constructing the node and edge functions and proving very tight bounds on their runtime. As an example, as noted by Reviewer Yrxd, the construction needs be done so carefully that it cannot even include simple-sounding steps like multiplying two small integers. Thus, the core technical content in our proofs is very new compared to MSS22.
>
> Lastly, simply having a constant depth computation graph does not guarantee a constant depth circuit. Whether this holds depends on a number of critical factors, including the kinds of gates allowed in the circuit (e.g., transformers are not in AC0, another fixed depth circuit class) and the kinds of operations performed at each graph node (e.g., our construction wouldn’t work if transformers used n-ary aggregation operations different from addition or multiplication).
>
> We hope this clarifies that our results are far from obvious or incremental, both in terms of their technical details and implications.
>
> ### Title
> Apologies for an ambiguous choice of words in the title. It is not our intent to claim that any FO(M) formula can be simulated by a transformer, which is why we don’t make this claim anywhere in the main body of the paper. Rather, we are claiming (only) that transformers can be simulated by FO(M) formulae. As you point out, we now see that the title could be read ambiguously to suggest the former. We will clarify it to avoid confusion, e.g.: *Transformers Can Be Translated to First-Order Logic with Majority Quantifiers*.
>
> ### Division Claim
> Our claim that transformers can be reduced to integer division does not rely on the claim that “integer division is reducible to TC0”. Rather, it relies on the fact that integer division is a *complete* problem for log-time uniform TC0, i.e., that every problem in log-time uniform TC0 can be reduced to integer division (Hesse, 2002).

---

### Official Review · Reviewer_Yrxd · 2022-11-04

**Confidence:** 4
**Correctness:** 1
**Technical Novelty And Significance:** 3
**Empirical Novelty And Significance:** Not applicable
**Recommendation:** 3

**Clarity, Quality, Novelty And Reproducibility:**

I give the authors credit for the quality of their presentation, making the bug easy to spot. In any case, without correctness, the other concerns are moot.

**Strength And Weaknesses:**

If correct, the claims in this paper are provocative and would be significant.

Unfortunately, there is a significant gap in the proof and I'm not sure that the claims are true.

In more detail, in the proposed DLOGTIME-uniform construction of transformer networks, the algorithm uses multiplication and division of the arguments, specifically bnode(n,i) = \lfloor i/bsize(n)\rfloor and bstart(n,i) = i * bsize(n). The text claims, "As bsize(n) = poly(n), these functions are reducible to arithmetic over O(log n)-bit integers, which can be implemented in O(log n) time." But, note that this claim is equivalent to arithmetic -- specifically here, multiplication and division -- of n-bit integers being implementable in O(n) time. I am confident that this is not known. A recent breakthrough by Harvey and van der Hoeven finally brought the complexity of multiplication down to O(n log n), but even this is not adequate for the needs of the construction. Since the DLOGTIME-uniformity of the circuits was the core of the argument, this seems to undermine the main claims of the paper.

Of course, it is not known that these operations require superlinear time, so the claims could be weakened to include the hypothesis that multiplication and division are linear-time computable. Since this hypothesis seems doubtful, it certainly undermines the significance (or surprise) of the claims -- we just have that one seemingly unlikely thing implies another.

**Summary Of The Paper:**

The paper claims that transformer networks can be represented by a constant-size formula of first-order logic with majority quantifiers; in turn, this means that the computation of such networks can be reduced to division, as division is complete for such problems.

**Summary Of The Review:**

The construction used to prove the main claim relies on improbably efficient integer arithmetic. It is a significant gap in the proof.

---

> ### Author Response · Authors · 2022-11-09
> **Clarification on Correctness**
>
> Thank you for your careful read of our paper! We appreciate that you caught a valid bug in Section 3.1. A small change to the proof suffices to squash the bug and make the results go through.
>
> The problem you point out in Section 3.1 is that we cannot guarantee a log-time Turing machine can implement the simple arithmetic to compute bnode and bstart, because they involve multiplying/dividing log(n) bit numbers by bsize (also a log(n)-bit number). We can get around this by simply padding bsize to be the smallest power of 2 greater than the maximum block size, which reduces multiplication and division to left and right shifting. These efficient operations can be computed in log(n) time on a number of size log(n) (i.e., linear time).
>
> The underlying assumption remains the same: there exists some polynomial bsize(n) computable in log(n) time. We then define bsize'(n) as the "rounded up" version of bsize(n), which can be computed in log(n) time by replacing all bits in bsize(n) with a 0 and prepending a 1. The new block-size bsize'(n) remains poly(n) because at most it is 2 * bsize(n). Thus, all the other results go through.
>
> We note that the same idea can be applied analogously to the column-uniform construction in the appendix (although this is not necessary to make our main result go through, just an alternate proof that is slightly more efficient in terms of the resulting circuit size).
>
> Thanks again for your careful read! We hope this addresses your main concern about correctness.

---

### Author Response · Authors · 2022-11-18
**Rebuttal version submitted**

Dear reviewers,

Thank you for your comments! We have submitted an updated rebuttal version addressing your reviews.

---

### Decision · Program_Chairs · 2023-01-20

**Decision:**

Reject

**Justification For Why Not Higher Score:**

see above

**Justification For Why Not Lower Score:**

N/A

**Metareview: Summary, Strengths And Weaknesses:**

Many technical details in this paper were unclear, identified both in the reviews and offline discussion. The authors kindly clarified a few key items (like the bug in the proof and how to fix it) but overall there is still too much lack of clarity for me to be comfortable accepting this paper. Especially since this is ICLR and not a CS theory conference, we run the risk of overlooking technical problems outside of our main area of expertise. Therefore clarity must be of the utmost quality before this can be accepted at ICLR.